# Using simulation to teach nursing students how to deal with a euthanasia request

**Dennis Demedts**[1,2]*, **Jürgen Magerman**[3,4,5], **Ellen Goossens**[6,7], **Sandra Tricas-Sauras**[1], **Johan Bilsen**[2], **Stefaan De Smet**[8], **Maaike Fobelets**[1,9,10]

1 Brussels Expertise Centre for Healthcare Innovation (BruCHI), Erasmus Brussels University of Applied Sciences and Arts, Brussels, Belgium, 2 Research group Mental Health and Wellbeing, Vrije Universiteit Brussel (VUB), Brussels, Belgium, 3 Research Collective EQUALITY, HOGENT University of Applied Sciences and Arts, Ghent, Belgium, 4 Department of Special Needs Education, HOGENT University of Applied Sciences and Arts, Ghent, Belgium, 5 Department of Special Needs Education, Ghent University, Ghent, Belgium, 6 School of Healthcare, HOGENT University of Applied Sciences and Arts, Ghent, Belgium, 7 Research Centre 360˚ Care and Well-being, HOGENT University of Applied Sciences and Arts, Ghent, Belgium, 8 Research Centre SUPRB, HOGENT University of Applied Sciences and Arts, Ghent, Belgium, 9 Biostatistics and Medical Informatics Research Group, Vrije Universiteit Brussel (VUB), Brussels, Belgium, 10 Department of Teacher Education, Vrije Universiteit Brussel (VUB), Brussels, Belgium

* dennis.demedts@vub.be

**Data Availability Statement:** All three files are available from the Mendeley database (https://doi.org/10.17632/9skh32kmbm.2).

## Abstract

Nursing students are confronted with euthanasia during their internship and certainly during their later career but they feel inadequately prepared in dealing with a euthanasia request. This study presents a simulation module focused on euthanasia and evaluates nursing students' perceptions after they have completed the simulation practice. The 'euthanasia module' consisted of a preparatory online learning module, a good-practice video, an in-vivo simulation scenario, and a debriefing session. The module's content was validated by four experts in end-of-life care. The module was completed by three groups of students from two different University Colleges (n = 17 in total). The students were divided into three groups: one with no previous simulation education experience, one with all students having previous experience, and another with a mix of experiences. After completing the entire module, each group had a discussion regarding their perceptions and expectations concerning this simulation module. Thematic content analysis was conducted on audio-recorded group interviews using NVIVO® software, involving initial open coding, transformation into specific themes and subthemes through axial coding, and defining core themes through selective coding, with data analysis continuing until data saturation was reached. Students generally found the online learning module valuable for background information, had mixed perceptions of the good-practice video, and appreciated the well-crafted scenarios with the taboo of euthanasia emerging during simulations, while the debriefing process was seen as enhancing clinical reasoning abilities. Students considered the simulation module a valuable addition to their education and nursing careers, expressing their satisfaction with the comprehensive coverage of the sensitive topic presented without sensationalism or taboos. This subject holds significance for nations with established euthanasia laws and those lacking such legal frameworks. The findings of this study can aid teachers in developing and accessing euthanasia simulation training

**Funding:** This study was funded by the Alliance Fund of Erasmus Brussels University of Applied Sciences and Arts and HOGENT University of Applied Sciences and Arts in the form of a grant to DD, JM, EG, SDS, and MF [HE_P43].

**Competing interests:** The authors have declared that no competing interests exist.

programs, contributing to broader education's emphasis on integrating euthanasia-related knowledge and skills.

## Introduction

Euthanasia raises many questions among healthcare professionals, including nurses. This is also the case with nursing students as they encounter this ethical dilemma during their internship, as well as during theoretical instructions. Although caring for and supporting patients close to death is part of the core tasks of the nursing profession, situations where patients themselves choose to shorten their agony through a self-selected death, such as euthanasia or physician-assisted suicide pose challenges. Euthanasia indicates the act of ending the life of a patient at its explicit request by means of lethal drugs administered by a physician [1, 2].

In Belgium, handling euthanasia requests follows a rigorous procedure, with the physician ultimately responsible. The Belgian Euthanasia Act [3] mandates consultation with the nursing team if they are involved in a patient's care with a pending euthanasia application. Nursing students may participate in preliminary decision-making. Exploring attitudes, skills, and roles, studies focussing on Belgian nursing students and graduated psychiatric nurses revealed a need for enhanced knowledge, skill development, and clear guidelines [4–8]. In response to the expressed need of nursing students and graduated psychiatric nurses for improved training in handling difficult and complex situations in euthanasia, a teaching module that utilizes simulation-based learning was developed as nursing students envision a clear role for themselves in the future but feel inadequately prepared for it. The simulation session covers various reasons for requesting euthanasia because our aim is to equip nursing students with the necessary skills to engage in discussions about euthanasia in general with their patients. Simulation learning has proven to be an effective educational approach for training both students and registered nurses, with nursing interventions based on simulation showing a significant educational impact [9]. In this educational strategy, students perform real professional activities in an authentic learning environment, including actors and patient simulators, under close supervision [9–12]. Simulation is widely used in nursing programs to teach psychiatry and mental health, besides technical skills such as cardiopulmonary resuscitation (CPR). It improves knowledge, empathy, confidence, communication, and reduces anxiety related to mental health needs. Standardized language and guides are recommended for simulation planning, with briefing, debriefing, and facilitation considered interconnected. Simulation education improves undergraduate nurses' skills, confidence, and knowledge in delivering mental health care in acute care settings, with evidence of transferability to clinical practice [11, 12]. Nursing educators identified creating a safe environment, facilitating student-centred learning, and promoting reflection as important factors in successfully implementing simulation exercises in undergraduate nursing education. Most of these nursing educators expressed a desire to include more simulations in their programs [10].

To the best of our knowledge, no studies are currently available that report on the use of simulation based nursing education to prepare students for dealing with a euthanasia request, despite simulation's effectiveness. Addressing this gap, a simulation module was developed, providing theoretical knowledge and scenarios involving patient requests for euthanasia. This study describes the development of a euthanasia simulation module and explores nursing students' perceptions in two University Colleges in Flanders, Belgium.

## Materials and methods

### Research setting

A total of 134 final-year nursing students of two Belgian University Colleges were invited via their learning platform to participate in this study. Students were offered the opportunity to participate in three different simulation sessions in the context of euthanasia. It was explicitly emphasised that participation was entirely voluntary. Participants were assigned to the simulation sessions according to their availability to join one of the training sessions in Brussels, the capital city of Belgium (one session) or Ghent, 3rd largest city of Belgium (two sessions), and to later share their experience with simulation education within the regular curriculum. Sessions were organised between November 2021 and April 2022.

### Research design

This qualitative study outlines the creation of a euthanasia simulation module and explores nursing students' perceptions of their simulation experiences in two University Colleges in Flanders, Belgium. Participants engaged in a preparatory online learning module before the simulation session, which included a good-practice video illustrating a nurse-patient conversation about euthanasia, followed by discussions on patient motives, available options, and role-play scenarios. Subsequently, a facilitated debriefing was conducted, and three group interviews were held to capture students' perceptions on the explored topic.

**Preparation of the online learning module.** Prior to the simulation session, participants followed a preparatory online learning module. In this evidence-based module, developed by the research team, students learned how to deal with questions about end-of-life, including euthanasia. This module consisted of a practical section on conversation techniques, focus points, and a juridical section to ensure that students can provide the correct information when necessary to the patient. This learning module was validated by four Belgian experts in end-of-life care. One of the experts is a psychologist actively working with end-of-life and the other three experts (a lawyer, a researcher, and a nurse) are involved in an end-of-life care organisation providing specific training and information on end-of-life care in the Belgian setting. The validation process involved first conducting individual interviews with each expert to identify the necessary content that should be included in the online learning module to ensure that students were adequately informed on both the legal framework and nursing-related aspects of the topic. Subsequently, the online learning module was presented to each expert, for their feedback. This feedback was incorporated to develop the final version of the learning module. The final version was approved by every expert.

**Simulation module.** The simulation module, based on the best and most recent evidence in the online learning module, starts by a good-practice video of a conversation between a nurse and a patient expressing a request for euthanasia because of unbearable mental suffering (UMS-euthanasia). In this good-practice video the caregiver adopts an attitude in which the request for euthanasia and life-ending in general is explored to discover 'the question behind the question'. After watching this video, students discuss the patient's expressed motives to request for euthanasia and the different available options, as well as interactions between the nurse and the patient that demonstrate good practice. The scenario of the good-practice video was validated by the four expert as well and a psychiatrist with experience in euthanasia.

The last part of the simulation module features three different simulation scenarios dealing with questions about life-end and euthanasia. Each simulation session included the enactment of all three scenarios. The presence of the perception of unbearable suffering is a defining characteristic of all scenarios. In each scenario, two students receive a description of their specific

role, simulating either a client or a nurse. The other students are observing using a minimal described scenario and are instructed to focus on verbal interaction and non-verbal responses. The scenarios are based on the most prevalent reasons to demanding euthanasia. One scenario was developed around cancer and one around polypathology as these are the two most common reasons. A third was UMS-euthanasia as this is legal in Belgium but also because this request is even more difficult to handle. As context, in Belgium, 2699 euthanasia acts were registered to the Federal Control and Evaluation Commission Euthanasia in 2021 [13]. The main categories of conditions that led to euthanasia requests were either malignant conditions (cancers-62.8%) or a combination of several serious and incurable conditions (polypathology-17.7%) which were beyond improvement and caused severe disabilities up to organ failure. UMS-euthanasia accounted for 0.9% of the total number of cases [13]. One University College had a simulated patient room equipped with a table and chairs, while the other college had a simulated home environment.

The role-play was followed by a 40-minute facilitated debriefing, which is twice as long as the scenario, as recommended by Dreifuerst [14]. Debriefing involves discussing the various aspects of the simulation, stimulating the participants' reflective thinking. In addition, it involves exploring, with the participants, their emotions and answering questions together as well as students' feedback [15]. The 'Debriefing for Meaningful Learning' (DML) method was chosen [14]. The simulation scenarios were recorded with a video camera, enhancing students' recall and feedback. During the debriefing, supervised by a skilled lecturer/facilitator, the role-playing students' experiences as well as the observers' comments are discussed and linked to the content of the theoretical module. After the transcription the video material were deleted permanently.

## Study population and sampling

Students received detailed information about the structure of the module two weeks prior to the training, including the purpose of the study; the online preparation expected of them; the course of the simulation exercise and the final discussion group with an interviewer. Neither the interviewer nor the simulation facilitator was involved in the students' regular curriculum. A simulation facilitator maximizes the use of simulation equipment and creates a context that is relevant and reliable, thereby promoting purposeful debriefing to facilitate learning effectively [16].

The first group consisted of five Ghent students in which none had experience with simulation education. The second group included six Brussels students, and in this case all of them had previous experience with simulation education. The third group encompassed six students, two from Brussels and four from Ghent. In this case only two students had a previous experience with simulation education.

## Data collection

Three group interviews were organised after the simulation sessions to map the students' perceptions on the topic explored. The moderation of sessions was carried out by two out of three researchers (DD, JM, EG), depending on the researcher's level of involvement in the curriculum. An interview guide with open questions was used to explore the participants' perceptions on the simulation experience in general, and each particular phase of the simulation module (see S1 Appendix).

The three simulation sessions and their respective discussion groups took place on separate days.

## Data analysis

The data underwent thematic content analysis, involving group interviews that were audio recorded and transcribed verbatim. Two researchers (DD and MF) familiarized themselves with the content and utilized NVIVO® software (QSR International Pty Ltd.) to generate initial codes (open coding). The initial ideas were then transformed into specific themes and sub-themes through axial coding. The research team engaged in extensive discussions to thoroughly analyse the data, ensuring consistency and integrity. Core themes emerging from open and axial coding were defined and named (selective coding). Finally, the results were organized and evaluated for coherence. The codes were categorized chronologically and deductively under different phases of the simulation. Data analysis continued until saturation, meaning that no new information about the topic emerged [17]. The group interviews are available on Mendeley [18].

## Ethical considerations

Ethical approval was obtained by the ethical committee of the University Hospital Brussels (B. U.N.1432020000138/I/U). A formal approved informed consent was obtained from all participants.

# Results

## Demographics

A total of 17 students from two University Colleges participated in three discussion groups. Most participants were female (n = 15, 88%) and the majority of students were under 25 years old (n = 14, 82%). About two-thirds had been confronted with euthanasia before (n = 11, 65%) either on a professional or on a personal level. About half of the students had experience in simulation training before (n = 8, 47%).

The topics discussed in the groups will be presented in chronological order, beginning with the motivation to participate in the euthanasia simulation module. This will be followed by the preparation phase, which includes the online learning module and the good-practice video. Next, the simulation itself, including the scenario and exercise, will be discussed, followed by the debriefing session. Finally, the perceived benefits of the simulation module in the students' learning process will be addressed.

## Euthanasia topic

During their theoretical programs, the topic euthanasia remained mainly limited to the theory basis without deepening on the implications at the practical level. Dealing with euthanasia is not a part of any of the participating University Colleges curricula. The limited focus on the topic in their regular courses resulted in an increased student interest for this practice-oriented module. Additionally, interest was raised because this is a sensitive topic concerning death where, despite the frequent incidence in practice, students appear to be searching for practical tools to deal with it.

'*It is a not much discussed topic too. And it has death as an ending actually. Then there is nothing left.*'

*Discussion group 2*

Furthermore, it was also underlined that euthanasia is a complex subject regulated by strict legislation, making healthcare workers vulnerable to possible prosecution when not following

these rules. Due to ignorance on a number of euthanasia related issues, there is a greater concern that something could go wrong during the administration of the drugs, but also during the preparatory process. The students indicate that families are normally present and overwhelmed by grief. This makes the relatives extra critical and sensitive towards the process.

*'I also think it is so sensitive because it is so complex, the legislation is so complex. . . if I were ever to perform that with a doctor and somehow something goes wrong, and the family sues me. . . Every now and then you hear it pass in the news. . .'*

*Discussion group 2*

Finally, it was stated that one cannot rely on internship experiences because there is a lot of ignorance among nurses and no unanimity in dealing with euthanasia among healthcare institutions. There is also still a taboo about euthanasia which means you cannot always discuss this topic with caregivers or lecturers.

*'. . .I haven't heard anyone talk about it at my internship yet. There is also little unanimity. Because Brussels does it one way and Leuven in another way and then there is no unanimity so that you can do it well and be open about it. . . You can't move forward with your feelings because it's such a taboo and there are much stricter rules there.'*

*Discussion group 2*

*'I have a home nursing internship now and there was someone who wanted euthanasia and she said that and I totally didn't know how to react.'*

*Discussion group 3*

To summarize, the students were motivated to participate due to their limited knowledge and lack of opportunity to practice the necessary skills, despite the importance of such knowledge and skills. Additionally, the students expressed a desire to reflect on the relevant legislation. Lastly, there was a need to discuss euthanasia as it remains a taboo topic.

## Preparation of the simulation

**Online learning module.** Students were asked to go through the online learning module in advance. Some students did not complete the task, while others prepared themselves appropriately. Few students also gathered additional information before joining the simulation session, such as visiting the LEIF-website. LEIF is an inclusive initiative that involves individuals and organizations dedicated to ensuring a dignified end of life for all individuals, with a primary focus on respecting the patient's will ([www.leif.be](http://www.leif.be)). Some students indicated they came unprepared to learn more from the simulation exercise.

*'I reviewed the PowerPoint and looked at the LEIF website to reiterate the legislation around it.'*

*Discussion group 1*

The preparatory online learning module was considered useful by participants. For some students it was an update of the legislation procedures, for others it was something totally new. Students appreciated the online learning module because it was brief and concise, as well as a useful tool as background information.

*'That legislation says you need two or three doctors. Underage is only for terminally ill. I didn't know all that'.*

*Discussion group 2*

As a point of reference, it should be noted that in Belgium, euthanasia is legal under strict conditions. It is permitted for minors experiencing unbearable terminal suffering, as well as adults experiencing unbearable non-terminal suffering. In these cases, three doctors are required to be involved. For adults experiencing unbearable terminal suffering, two doctors are required [3].

**Good-practice video.** The simulation started with a "good-practice" video. This video was generally considered as positive but was perceived differently by students. The good-practice video provided guidance on how to conduct a conversation on the topic of euthanasia, which focus and the type of questions to ask during such an exchange and how such a conversation can proceed.

*'. . .And I liked that we could pay attention to those things. Like that listening attitude and citing euthanasia early. That we could then pay attention to. . .'*

*Discussion group 3*

There was no unanimity on whether the video should be shown before, or after the simulation exercise. Some students felt the video would guide them into a too narrow direction, while others needed that guidance. However, engaging in group discussions about the video was deemed essential as it not only provides fresh perspectives but also helps in validating one's own approach and ideas.

*'Maybe you just learn more from it than if you don't show something beforehand, because then you're just put into it and then you follow your own feelings to test: how do you really deal with that instead of having that example.'*

*Discussion group 2*

**Simulation scenario.** Following the viewing of the 'good-practice video', students were presented with various simulation scenarios prior to beginning their practice. Students commented that the scenarios were well-crafted, providing enough (medical background) information while remaining highly recognizable. The scenarios were detailed, yet allowed students some flexibility to put their own spin on the situation. Some students appreciated this flexibility, while others requested more detailed guidance. Sufficient background information was given about the patient, though some students felt that more emphasis could be placed on the relationship of trust with the patient. Nonetheless, the scenarios provided sufficient information for each role, enabling students to empathize fully.

*'They are recognisable situations, which helps for empathy.'*

*Discussion group 1*

*'I like that we were given the context around it. . .. Here you do have "kidney cancer", "parents apart". . ., you can prepare for the conversation more than in real life.'*

*Discussion group 3*

**Simulation exercise.** About half of the students had previously completed a simulation exercise. However, the previous simulation sessions had focused on practical-executive nursing skills, such as cardiopulmonary resuscitation (CPR) rather than communication skills in a well-defined situation such as a euthanasia request. For the students for whom this simulation was a first one, the exercise was not so easy. Additionally, the more experienced students also indicated that this simulation module was not suitable for inexperienced students in simulation. The students with a previous experience, felt comfortable knowing how simulation works and what is expected. This allowed them to focus more on the content. Empathizing with their assigned role was challenging for both experienced and inexperienced students. It did support the students to act more authentic knowing there is no result assessment linked to this training.

*'I found it difficult to find myself in my role. I didn't think it was going to be so difficult, but I found it instructive.'*

*Discussion group 3*

*'From being in your role you do forget after a while that you are in a role-play, but you keep realising that it is not real.'*

*Discussion group 1*

Also, students themselves experienced the taboo about euthanasia once they stepped in their roles during simulation. Both in the case of their role as a nurse and the patient's role. They expressed that it was not always easy to talk straight about euthanasia. Despite the awareness that such conversations are a crucial part of their future job, participants acknowledged that it did not always feel natural to engage in them. When considering the patient's role, it sometimes felt strange for them "to request euthanasia" as such when they did not feel they could ever ask for it themselves.

*'. . .We then got the scenario and when you hear, okay, I understand they want that. But once you have to explain it all like that and get to the point of saying, euthanasia, is that how it is now, then I don't find it so easy.'*

*Discussion group 2*

According to the students' feedback, the size of the group played a significant role in their experience. The students noted that smaller groups (consisting of 5–6 students) were ideal as they allowed for better communication without any inhibitions and reduced the pressure to perform. Moreover, the absence of any grading system was also appreciated. Simulating with fellow students gives less trepidation although there is no unanimity on who should take on the role of the patient. On the one hand, it is instructive for a student to take on the role of the patient because in this way, they learn to understand the patient, but on the other hand it can be an added value for the lecturer or an actor to take on the role too because the conversation can be more guided and thus be more in-depth. Students recognize that having a real patient who genuinely contemplates euthanasia would be the ideal scenario for fostering deeper empathy. However, they also acknowledge that this is often unrealistic. Students firmly express their disapproval of using simulation mannequins under any circumstances.

*But I think teachers actually play their roles well. It is good that we can then indeed also go deeper into the conversation, but of course, it might be threatening for some students.'*

*Discussion group 2*

*'You talk to that, but you know that's a mannequin. . .. If it's a real person, then you pay attention to the face etc. . .'*

*Discussion group 3*

Finally, students indicate that a realistic environment should be created. This needs to be a simulated environment within the walls of the University College. This can be a classic simulated hospital room, as well as a simulated home environment.

*'It's a good thing that that room had a living room feel. Otherwise, it was very fake.'*

*Discussion group 3*

**Debriefing.** The simulation scenarios were followed by a debriefing session, in which each part of the simulation, as well as feelings and emotions that emerged were discussed. Students find debriefing valuable and essential. They indicated that they were somewhat apprehensive at the beginning of the debriefing session in case they would have said wrong things during their exercises. They also express that their intense focus during the simulations made it challenging to envision how they would have actually responded in real-life situations. Nevertheless, debriefing remains a very stressful event for some of the students.

*'When I came here, I had some anxiety because I thought I had said something totally wrong.'*

*Discussion group 1*

*'I did have stress; you are not judged with points, but you are looked at. Whereas in a healthcare setting, you just do your thing with your patient and it's much more spontaneous. And not now, I'm a stressed chicken too.'*

*Discussion group 3*

The majority of the students who participated in the simulation were accustomed to debriefing sessions that focused on the principles of Crew Resource Management (CRM) [19]. However, for this particular type of simulation, the decision was made to use the more reflective approach 'Debriefing for Meaningful Learning' (DML) [14] that gave attention to the main topic of the simulation, allowed space for feedback, and considered the emotions of the participants. The students recognized the value of this approach and indicated that using CRM-principles for debriefing was less suitable for this type of simulation.

*'By reflecting with the group afterwards, you also learn from other people. In real life, you can't just go and ask a colleague "So, did I do it right", they don't know anything about it. So, reflecting with a group helps a lot with that.'*

*Discussion group 3*

*'I like that reflecting better, half the CRM-principles are not covered.'*

*Discussion group 1*

Furthermore, students indicated that they learned from each other through the debriefing sessions. This learning aspect is a great added value for them. Discussing it with each other

also helps in questioning and possibly adjusting their own views. After all, it is not so easy to get feedback on their communicative actions in the work field, for instance in this case, due to the sensitivity of the chosen topic.

*'You are going to pick up things like that was a good sentence or maybe you could have said it that way. You're going to remember it now. . . Because everyone also said: oh, I don't actually remember what I said at that moment. And then it's good to bring it up again.'*

*Discussion group 2*

*'Different perspectives, learning from each other. Everyone has their own way of working and experience of internship. That way you don't sit and push your own vision.'*

*Discussion group 3*

Finally, students indicated that being filmed during their simulation was not a problem. This allows them to revisit their own actions during debriefing. Furthermore, it was clearly stated that there was no assessment linked to this simulation training. In particular, they suggested that the fact that there was not an assessment as such, decreased the stress.

*'. . . you get feedback, but you totally don't remember how you behaved. Then when you see yourself doing it, you can look at yourself more critically and take more with you.'*

*Discussion group 1*

**Added value.** Students indicate that this simulation practice is a substantial added value to their nursing education. Only theory-led education is not enough to gain a proper understanding towards such a sensitive topic as euthanasia.

*'. . .In the first year, you only learn about patients' rights. So, it's really interesting to do a simulation on this.'*

*Group discussion 3*

Students indicated that this simulation exercise added value to their future nursing practice. It was also beneficial for them to explore different cases, to identify several reasons for demanding euthanasia, and to consider different perspectives during this exercise. This allows them to be better prepared to respond to their future practice as nurses in the context of euthanasia. If confronted with a euthanasia question in the future, students believed that they would revisit their experience during this exercise.

*'I would think back to it anyway of how to do this now, what can I say. I think it would be a bit easier then.'*

*Discussion group 1*

*'We now know what to say or how to answer and others may not know. So, I do find it interesting and I am glad we came here. Yes, fulfilled.'*

*Discussion group 2*

A few students expressed that this was just the start. They gained insights on how to listen to and address euthanasia-related issues from a particular role (either nurse or patient). Some

of them expressed a desire to repeat the module, but from the other role. However, there was no consensus among the students regarding this matter.

*'. . .We have now all had a taste in this session: we can be patients or nurses and what can we say? That you then do the second round, then you have actually tasted both roles once. Everyone is the core patient, and everyone is the core nurse. . .'*

*Discussion group 2*

## Discussion

This study describes the design of a simulation module about euthanasia, as well as the perceptions of the students who participated in this new simulation module. The topic of euthanasia is highly relevant to the Belgian setting because a euthanasia law exists since 2002 [3] although not all professionals feel prepared to deal with such a topic [4, 6–8]. Consequently, nursing students may be confronted with euthanasia, during their internship but undoubtedly will face this issue during their future career.

In this research, students reported that euthanasia is a subject that is very tangentially discussed during their programs, although more than half of the students have already been confronted with euthanasia. Due to inadequate preparation and the taboo still associated with euthanasia students do not feel skilled enough when facing such situations during their internships. Thus, this simulation module aimed to prepare students to in-depth explore a euthanasia question. In the development of this simulation module, three key components were incorporated: a preparatory phase consisting of an online PowerPoint, a good-practice video, and a simulation scenario; a simulation role-play; and a debriefing session. This simulation exercise was perceived as beneficial and will add value in their future nursing practices.

Both in this study and previous research on the topic [5, 6], more than half of the students reported having been confronted to euthanasia. This may have been during internships or their personal lives. Since the Belgian euthanasia law came into force, an annual gradual increase in the number of registered euthanasia acts is noticed, resulting in 2699 registrations in 2021 [13]. Despite nurses having a key role in the process of euthanasia (clearance, preparation, administration, aftercare), their role is not yet clearly defined by legislation, leaving nurses in a grey zone. This is also indicated by the final-year undergraduate students in this study. There is a lack of clarity on what is legal and what is not. For this reason, our preparatory online learning module explained the legislation in all its components, as well as possible scenarios (including the grey zones) in which those involved may find themselves. Different roles were described for the nurse across the different phases leading up to euthanasia. These phases were: initial care and accompaniment (Receive the euthanasia request; Prior accompaniment); evaluation of the patient's condition (Application management; Application analysis; Decision making); procedure for assistance in dying (Drug preparation; Drug administration; Postmortem care); debriefing and notification (Self-care; Participation and evaluation) [20, 21]. Within this simulation module, the focus was on the first phase: initial care and accompaniment. This phase contains the receipt of the request, advising to also discuss this request with a physician, inquiring whether the request may be shared with the multidisciplinary team, indicating that a request can always be withdrawn, as well as informing about palliative care. In addition, this phase includes listening, understanding, responding, and supporting the patient and significant others. This includes correctly displaying information or referring to appropriate information channels [20, 21]. Those issues were therefore addressed in the online learning module because it is very important to further explore the euthanasia question so that

there is clarity on whether or not this wish is lived through. After all, a euthanasia question can also be an expression of another wish or thought. Because the question needs to be considered in depth before moving on to the other phases, we focused on this phase in this simulation. To achieve deeper understanding, students should have a foundational theoretical knowledge. There was unanimity of the added value of the online learning module and it was considered enlightening. For some students it was a refresher, for others simply innovative. Students revealed that this topic is underexposed in the nursing program. Despite not being a mandatory part of the nursing curriculum and therefore not always having a strong presence, nurses are occasionally required to provide palliative care in addition to their curative work [22]. Besides the theoretical part, the communicative aspect was also explained in the good-practice video. In it, students are instructed how to explore the meaning, and offer information about the possibility of euthanasia, but take a neutral stance so that they refrain from encouraging, suggesting, recommending, advising or inducing euthanasia [20]. Although the video was seen as useful, there was no consensus on when the video should be shown. The more inexperienced students were more likely to indicate that it was of useful guidance, the more experienced students fear that they will refer back to the video too much and therefore wish to see the video only after the simulation. As previously mentioned, simulation experience is beneficial when the simulation centres on sensitive topics and communication skills, such as euthanasia and suicide [23]. The module on euthanasia was deemed an important learning experience for students, which is why it was offered to final-year students. This allowed them to acquire adequate theoretical background, internship experience, reflective skills, and simulation experience. Interestingly, students indicate that even from their simulated position in the scenario, it was not always easy to talk directly about euthanasia. From their roles, they sometimes felt some trepidation. Experiencing this simulation forces students to adopt a reflective attitude towards their own attitude towards euthanasia. This in itself is also a positive achievement of the simulation module. Demedts et al. [5] suggest that more attention should be paid to a reflective attitude around euthanasia in order to better prepare students for their future nursing role. Students indicate that the groups should not be too large. Hence, three scenarios were provided for the simulation scenario with one patient and one nurse in each case. This allowed each student to take up at least one role, as well as observing the others at least once.

Debriefing, including feedback and reflection, are essential parts of simulation education that can provide opportunities to improve clinical performance. Debriefing, gives students the opportunity to reflect on the experience gained but also to revise certain actions by exchanging feedback. Debriefing further develops clinical reasoning but also judgement [24]. For this reason, the 'Debriefing for Meaningful Learning' (DML) method [14] was chosen. DML is a systematic process for debriefing in which teachers and students reflect together and generate new meanings from the simulation experience. Although appreciated by the students, this form of debriefing is time- and intensive because it requires an instructor-led debriefing where the formally trained instructor leads the debriefing [25]. Students clearly indicated that this reflective format was preferable to their usual method of reflection using the CRM principles. Furthermore, sessions were recorded which allowed us to review particular actions with the students. Niu et al. [26] indicated that video-assisted debriefing improved students' experiences and learning outcomes related to critical thinking skills by reducing bias when evaluating scenarios. Forbes et al. [27] do indicate that students may feel pressured to perform for the video. For this reason, the instructor should create a supportive and respectful environment so that they feel safe. Some students did indicate that they felt still stressed but in itself this is not a problem as long as the stress does not prevent their actions. After all, during a workplace situation, they will experience similar stress and so they should be aware of the stress factor and

its influence. Students found the non-judgemental nature of the simulation reassuring and manageable in terms of stress. Partly by not awarding grades, a safe environment was created. As a result, filming their simulation exercise was not a problem. Because the cameras were hidden away in the simulation room, some students even forgot they were being filmed.

Because of the reasons mentioned above, students found this a very instructive exercise. In previous research, students indicated that they did not feel skilled enough in dealing with euthanasia [6, 7]. This simulation module offers a partial solution here. Students in the study indicated that only theoretical knowledge is not enough and thus see the usefulness of this simulation. However, theoretical education and simulation training are not enough to achieve a solid foundation in dealing with a euthanasia request. To understand one's own attitude towards euthanasia, multidisciplinary debates can be useful [5, 7]. However, this simulation module is a step in learning to better cope with a euthanasia request. Students expressed that when confronted, they can take away the experiences from the simulation module and that these will help them further.

### Strengths and limitations

This study reckons some limits and strengths. First, there is the innovative aspect. To our knowledge, this is the first study to describe a simulation module that explicitly addresses euthanasia requests and how to deal with it. There have been previous publications about simulation addressing palliative care and end-of-life communication [28] but not specifically to euthanasia. Regarding the simulation module in itself, some strengths can be identified such as the participation of two University Colleges from two different Belgian provinces. By limiting the groups to six students, all participants were able to assume an active role during the simulation session, ensuring their active engagement rather than solely observatory participation. Additionally, there was not only a group from each respective institution but also a mixed group comprising students from both University Colleges. Besides the diversity of students, another strength is that the simulation module was validated by four external experts. We consider it a strength that we conducted group discussions after going through the full simulation instead of using questionnaires. This method gave us an in-depth insight that would not be obtained through a survey. Besides those strengths, there are also some weaknesses. Participation in the simulation training was voluntary, so students who participated were possibly already interested in this particular topic. Furthermore, it was evident that not all students were equally prepared for this exercise. Despite the availability of an online learning module, not every student had completed it. Consequently, some students expressed theoretical gaps that hindered their ability to fully benefit from the simulation exercise. Concerning the role of the patient during simulation, no unanimity was found among the students on who should take the patients' role. We opted for the students themselves so they could experience the feelings of the patient. Possibly, using simulation patients could provide an even deeper experience but we did not test this option in our module. Finally, only nursing students were included in this study, so our conclusions can only be applied to this particular group and not to other healthcare students who may also encounter euthanasia-related issues.

### Conclusion

This article describes the development of novel simulation module on how to deal with a euthanasia request. This work also describes students' feedback as well as perceptions on it. This module is designed for nursing programs but could be envisaged for other professionals. The simulation module has added value to the students' education and future careers as

nurses. Their positive feedback underscores the comprehensive coverage of this sensitive topic. The need for enhancing knowledge, skills, and clear guidelines in the domain of euthanasia-related education becomes apparent from the students' perceptions. The results of this study can support teachers in designing and evaluating similar simulation-based modules into their nursing programs, providing a structured and practical approach to addressing the complexities of euthanasia. It is important to tailor such modules to different geographical and legal contexts, ensuring their relevance and effectiveness in diverse settings. This research contributes to the broader educational landscape by emphasizing the integration of euthanasia-related knowledge and skills, ultimately better preparing nursing students for the challenges they may encounter in their future careers.

## Supporting information

**S1 Appendix. Interview guide with open questions to explore the participants' perceptions on the simulation experience in general, and each particular phase of the simulation module.**
(DOCX)

## Acknowledgments

We would like to thank the professionals from the field for their help and support in the validation of this simulation module.

## Author Contributions

**Conceptualization:** Dennis Demedts, Jürgen Magerman, Ellen Goossens, Johan Bilsen, Stefaan De Smet, Maaike Fobelets.

**Data curation:** Dennis Demedts, Jürgen Magerman, Ellen Goossens, Maaike Fobelets.

**Formal analysis:** Dennis Demedts, Maaike Fobelets.

**Funding acquisition:** Dennis Demedts, Jürgen Magerman, Ellen Goossens, Stefaan De Smet, Maaike Fobelets.

**Investigation:** Dennis Demedts, Jürgen Magerman, Ellen Goossens, Maaike Fobelets.

**Methodology:** Dennis Demedts, Jürgen Magerman, Johan Bilsen, Stefaan De Smet, Maaike Fobelets.

**Project administration:** Dennis Demedts.

**Resources:** Dennis Demedts.

**Supervision:** Johan Bilsen, Stefaan De Smet, Maaike Fobelets.

**Validation:** Dennis Demedts, Ellen Goossens.

**Visualization:** Dennis Demedts, Ellen Goossens.

**Writing – original draft:** Dennis Demedts, Jürgen Magerman, Ellen Goossens, Maaike Fobelets.

**Writing – review & editing:** Dennis Demedts, Jürgen Magerman, Ellen Goossens, Sandra Tricas-Sauras, Johan Bilsen, Stefaan De Smet, Maaike Fobelets.

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
