## [Decision Letter · Decision Letter 0]

11 Sep 2023

PONE-D-23-21618Using simulation to teach nursing students how to deal with a euthanasia requestPLOS ONE

Dear Dr. Demedts,

Thank you for submitting your manuscript to PLOS ONE. After careful consideration, we feel that it has merit but does not fully meet PLOS ONE’s publication criteria as it currently stands. Therefore, we invite you to submit a revised version of the manuscript that addresses the points raised during the review process.

We look forward to receiving your revised manuscript.

Kind regards,

Elif Ulutaş Deniz

Academic Editor

PLOS ONE

Dear Dennis Demedts, MSc,

Manuscript ID PONE-D-23-21618 entitled "Using simulation to teach nursing students how to deal with a euthanasia request " which you submitted to PLOS ONE, has been reviewed. The comments of the reviewer(s) are included at the bottom of this letter.

The reviewer(s) have recommended publication, but also suggest some revisions to your manuscript. Therefore, I invite you to respond to the reviewer(s)' comments and revise your manuscript.

Best regards,

Elif Ulutaş Deniz

Comments to the Authors from Academic Editor

Dear Authors,

The reviewers have provided their comments and recommendations. Based on their feedback, you are encouraged to revise the manuscript, addressing and responding to each comment specifically.

Best regards,

Comments to the Authors from Reviewer 1:

This qualitative study addresses an important topic and thoroughly engages with a simulation module on the topic of euthanasia, as well as interviews with nursing students who had completed this simulation module. The methods and statistics are reasonable, and the conclusions drawn are valid. However, some statements need to be clearer, and the language should be edited for publication consideration. Here are some specific comments for each section.

Format:

1. Please correct the in-text citations. Citations need to be included in the sentence (i.e., citations with brackets before punctuation), and blank spaces are needed between characters and brackets.

Title:

1. Suggest mentioning the “students’ perceptions towards simulation” in the title since it is a key aspect of the results.

Abstract:

1. Suggest adding more details about the methods, such as participants and data analysis.

2. In the results section, please add a sentence to describe the sample characteristics.

3. Please make sure that the abstract does not exceed 300 words.

4. Some sentences are a bit confusing, such as “Some students find it a useful tool, others find it too guiding.” and “However, specific nuances should be provided and therefore the module adapted to varied geographical contexts.” Please express these idea succinctly and clearly.

Introduction:

1. The author provides much information on simulation that does not show a close association with the research question. Please revise the introduction and demonstrate the dilemmas regarding euthanasia encountered by nursing students thoroughly to make the gap clear and the research rationale sound.

2. From lines 58 to 59, the author provides the definition of physician-assisted suicide which is not closely associated with the research question.

3. In lines 66 and lines 73, the author emphasizes unbearable mental suffering (UMS-euthanasia), yet the intervention in this study is not solely for UMS-euthanasia.

Methods:

1. Please provide more information about the Procedure and Tools section. For example, should all sessions be completed within a day, or are they divided into different days? In the Simulation Module section, should each group experience three scenarios, or does each group choose one?

2. Please introduce data analysis in a separate section rather than combining it in the “Data Collection” section.

Results:

1. Please provide a table displaying the sociodemographic data of the participants.

2. Suggest reporting the results following the typical structure of qualitative research, generalizing by theme and sub-theme, rather than by intervention implementation session.

Discussion:

1. Suggest moving strengths and limitations to the end of the Discussion and titling this section “Strengths and limitations”.

2. Lines 419 to lines 420 mention the sample size, which consists of 17 students from two universities. It is unclear whether this is considered a strength of the study, clarification is needed.

Comments to the Authors from Reviewer 2:

Abstract

-The abstract lacks the research methods which were used to conduct the study.

-The authors also did not write any recommendations yet they are important for readers to see the significance of the study.

Introduction

-The topics being studied was well introduce and the purpose of the study was described well.

-In lines 65 to 73, the authors cited a previous study and that description was too detailed. It was going to be better if this was a literature review, but since this an introduction, the authors should make their own argument and then support it with previous literature.

-The in-text referencing is wrongly done, It would never be before a sentence or after a full-stop. Please read about this.

Methodology

-Methodology is this study was done poorly. Please discuss the methodology under these subtopics or at least make sure vital information from these aspects is not omitted in your article:

-Research setting

-Research design

-Study population and sampling

-Data collection

-Data analysis

-Ethical considerations

Results and Discussion

-This is fine.

Conclusion

-Your conclusion talks mostly on the significance of the study. Please write again about your results like you did on the abstract and the add recommendation.

Reviewers' comments:

Reviewer's Responses to Questions

**Comments to the Author**

1. Is the manuscript technically sound, and do the data support the conclusions?

Reviewer #1: Yes

Reviewer #2: Yes

2. Has the statistical analysis been performed appropriately and rigorously? 

Reviewer #1: Yes

Reviewer #2: N/A

3. Have the authors made all data underlying the findings in their manuscript fully available?

Reviewer #1: No

Reviewer #2: Yes

4. Is the manuscript presented in an intelligible fashion and written in standard English?

Reviewer #1: Yes

Reviewer #2: Yes

5. Review Comments to the Author

Reviewer #1: This qualitative study addresses an important topic and thoroughly engages with a simulation module on the topic of euthanasia, as well as interviews with nursing students who had completed this simulation module. The methods and statistics are reasonable, and the conclusions drawn are valid. However, some statements need to be clearer, and the language should be edited for publication consideration. Here are some specific comments for each section.

Format:

1. Please correct the in-text citations. Citations need to be included in the sentence (i.e., citations with brackets before punctuation), and blank spaces are needed between characters and brackets.

Title:

1. Suggest mentioning the “students’ perceptions towards simulation” in the title since it is a key aspect of the results.

Abstract:

1. Suggest adding more details about the methods, such as participants and data analysis.

2. In the results section, please add a sentence to describe the sample characteristics.

3. Please make sure that the abstract does not exceed 300 words.

4. Some sentences are a bit confusing, such as “Some students find it a useful tool, others find it too guiding.” and “However, specific nuances should be provided and therefore the module adapted to varied geographical contexts.” Please express these idea succinctly and clearly.

Introduction:

1. The author provides much information on simulation that does not show a close association with the research question. Please revise the introduction and demonstrate the dilemmas regarding euthanasia encountered by nursing students thoroughly to make the gap clear and the research rationale sound.

2. From lines 58 to 59, the author provides the definition of physician-assisted suicide which is not closely associated with the research question.

3. In lines 66 and lines 73, the author emphasizes unbearable mental suffering (UMS-euthanasia), yet the intervention in this study is not solely for UMS-euthanasia.

Methods:

1. Please provide more information about the Procedure and Tools section. For example, should all sessions be completed within a day, or are they divided into different days? In the Simulation Module section, should each group experience three scenarios, or does each group choose one?

2. Please introduce data analysis in a separate section rather than combining it in the “Data Collection” section.

Results:

1. Please provide a table displaying the sociodemographic data of the participants.

2. Suggest reporting the results following the typical structure of qualitative research, generalizing by theme and sub-theme, rather than by intervention implementation session.

Discussion:

1. Suggest moving strengths and limitations to the end of the Discussion and titling this section “Strengths and limitations”.

2. Lines 419 to lines 420 mention the sample size, which consists of 17 students from two universities. It is unclear whether this is considered a strength of the study, clarification is needed.

Reviewer #2: Abstract

-The abstract lacks the research methods which were used to conduct the study.

-The authors also did not write any recommendations yet they are important for readers to see the significance of the study.

Introduction

-The topics being studied was well introduce and the purpose of the study was described well.

-In lines 65 to 73, the authors cited a previous study and that description was too detailed. It was going to be better if this was a literature review, but since this an introduction, the authors should make their own argument and then support it with previous literature.

-The in-text referencing is wrongly done, It would never be before a sentence or after a full-stop. Please read about this.

Methodology

-Methodology is this study was done poorly. Please discuss the methodology under these subtopics or at least make sure vital information from these aspects is not omitted in your article:

-Research setting

-Research design

-Study population and sampling

-Data collection

-Data analysis

-Ethical considerations

Results and Discussion

-This is fine.

Conclusion

-Your conclusion talks mostly on the significance of the study. Please write again about your results like you did on the abstract and the add recommendation.

6. PLOS authors have the option to publish the peer review history of their article (what does this mean?). If published, this will include your full peer review and any attached files.

Reviewer #1: No

Reviewer #2: No

---

## [Author Response · Author response to Decision Letter 0]

13 Nov 2023

We would like to express our sincere gratitude to the editor and the two reviewers who took the time to review our article and provide valuable feedback. We have endeavoured to carefully consider and incorporate the feedback received. Thanks to this targeted and constructive input, we feel that the article has been elevated to a higher standard. We extend our heartfelt thanks for this.

Furthermore, we have thoroughly reviewed the guidelines and hope that we now comply with the document naming conventions.

We adress all comments in a seperate rebuttal letter as response to the reviewers.

---

## [Decision Letter · Decision Letter 1]

16 Jan 2024

PONE-D-23-21618R1Using simulation to teach nursing students how to deal with a euthanasia requestPLOS ONE 

Dear Dr. Demedts,

Thank you for submitting your manuscript to PLOS ONE. After careful consideration, we feel that it has merit but does not fully meet PLOS ONE’s publication criteria as it currently stands. Therefore, we invite you to submit a revised version of the manuscript that addresses the points raised during the review process.

We look forward to receiving your revised manuscript.

Kind regards,

Elif Ulutaş Deniz

Academic Editor

PLOS ONE

Journal Requirements:

Additional Editor Comments:

Dear Dennis Demedts,

Manuscript PONE-D-23-21618R1 titled "Using simulation to teach nursing students how to deal with a euthanasia request" which you submitted to PLOS ONE, has been reviewed. The comments of the reviewer(s) are included at the bottom of this letter.

The reviewers suggest some minor revisions to your manuscript. Therefore, I invite you to respond to the reviewer(s)' comments and revise your manuscript.

Best regards,

Elif Ulutaş Deniz

Reviewers' comments:

Reviewer's Responses to Questions

**Comments to the Author**

1. If the authors have adequately addressed your comments raised in a previous round of review and you feel that this manuscript is now acceptable for publication, you may indicate that here to bypass the “Comments to the Author” section, enter your conflict of interest statement in the “Confidential to Editor” section, and submit your "Accept" recommendation.

Reviewer #1: All comments have been addressed

Reviewer #2: All comments have been addressed

2. Is the manuscript technically sound, and do the data support the conclusions?

Reviewer #1: Yes

Reviewer #2: Yes

3. Has the statistical analysis been performed appropriately and rigorously? 

Reviewer #1: N/A

Reviewer #2: Yes

4. Have the authors made all data underlying the findings in their manuscript fully available?

Reviewer #1: Yes

Reviewer #2: Yes

5. Is the manuscript presented in an intelligible fashion and written in standard English?

Reviewer #1: Yes

Reviewer #2: Yes

6. Review Comments to the Author

Reviewer #1: This qualitative study addresses an important topic and thoroughly engages with a simulation module on euthanasia, in addition to conducting interviews with nursing students who have completed this simulation module. The revised manuscript shows significant improvement, presenting a more coherent narrative. However, some minor points still require clarification.

Format:

1. The reference 4 to 8 are cited repeatedly in line 63.

2. There is a separate paragraph just one sentence in line 84. Please make sure is it a mistake.

Reviewer #2: The article is satisfactory. All the required specifications were met and the researchers did the suggested corrections.

7. PLOS authors have the option to publish the peer review history of their article (what does this mean?). If published, this will include your full peer review and any attached files.

Reviewer #1: No

Reviewer #2: No

---

## [Author Response · Author response to Decision Letter 1]

16 Jan 2024

We address all comments in a separated letter.

---

## [Decision Letter · Decision Letter 2]

5 Feb 2024

Using simulation to teach nursing students how to deal with a euthanasia request

PONE-D-23-21618R2

Dear Dr. Demedts,

We’re pleased to inform you that your manuscript has been judged scientifically suitable for publication and will be formally accepted for publication once it meets all outstanding technical requirements.

Kind regards,

Elif Ulutaş Deniz

Academic Editor

PLOS ONE

Additional Editor Comments (optional):

Reviewers' comments:

Reviewer's Responses to Questions

**Comments to the Author**

1. If the authors have adequately addressed your comments raised in a previous round of review and you feel that this manuscript is now acceptable for publication, you may indicate that here to bypass the “Comments to the Author” section, enter your conflict of interest statement in the “Confidential to Editor” section, and submit your "Accept" recommendation.

Reviewer #1: All comments have been addressed

2. Is the manuscript technically sound, and do the data support the conclusions?

Reviewer #1: Yes

3. Has the statistical analysis been performed appropriately and rigorously? 

Reviewer #1: N/A

4. Have the authors made all data underlying the findings in their manuscript fully available?

Reviewer #1: Yes

5. Is the manuscript presented in an intelligible fashion and written in standard English?

Reviewer #1: Yes

6. Review Comments to the Author

Reviewer #1: The paper meets expectations, with all necessary criteria fulfilled, and the researchers have incorporated the suggested revision.

7. PLOS authors have the option to publish the peer review history of their article (what does this mean?). If published, this will include your full peer review and any attached files.

Reviewer #1: No

---

## [Editor Report · Acceptance letter]

19 Mar 2024

PONE-D-23-21618R2 

PLOS ONE

Dear Dr. Demedts, 

I'm pleased to inform you that your manuscript has been deemed suitable for publication in PLOS ONE. Congratulations! Your manuscript is now being handed over to our production team.

Kind regards, 

on behalf of

Dr. Elif Ulutaş Deniz 

Academic Editor

PLOS ONE